# Humor in parenting: Does it have a role?

Lucy Emery[1], Anne Libera[2], Erik Lehman[3], Benjamin H. Levi[4]*

1 Penn State College of Medicine, Hershey, Pennsylvania, United States of America, 2 Theatre Department, Columbia College, Chicago, IL, United States of America, 3 Department of Public Health Sciences, Penn State College of Medicine, Hershey, Pennsylvania, United States of America, 4 Departments of Humanities & Pediatrics, Penn State College of Medicine, Hershey, Pennsylvania, United States of America

* bhlevi@psu.edu

**Data Availability Statement:** The data that support the findings of this study are openly available on Figshare http://doi.org/10.6084/m9.figshare.20404107, reference number 20404107.

**Funding:** The author(s) received no specific funding for this work.

## Abstract

### Background

Despite the widespread use of humor in social interactions and the considerable literature on humor in multiple fields of study, the use of humor in parenting has received very little formal study. The purpose of this pilot study was to gather preliminary data on the use of humor in the raising of children.

### Materials and methods

We developed and administered a 10-item survey to measure people's experiences being raised with humor and their views regarding humor as a parenting tool. Responses were aggregated into Disagree, Indeterminate, and Agree, and analyzed using standard statistical methods.

### Results

Respondents (n = 312) predominantly identified as male (63.6%) and white (76.6%) and were (by selection) between the ages of 18–45 years old. The majority of participants reported that they: were raised by people who used humor in their parenting (55.2%); believe humor can be an effective parenting tool (71.8%) and in that capacity has more potential benefit than harm (63.3%); either use (or plan to use) humor in parenting their own children (61.8%); and would value a course on how to utilize humor in parenting (69.7%). Significant correlations were found between the use of humor and both i) the quality of respondents' relationships with their parents and ii) assessments of how good a job their parents had done.

### Conclusions

In this pilot study, respondents of childbearing/rearing age reported positive views about humor as a parenting tool.

**Competing interests:** The authors have declared that no competing interests exist.

## Introduction

Parenting can be challenging, especially considering the wide array of responsibilities and other stressors most parents have in their lives. [*Parent* will be used here as a general term to describe any adult who has a primary role in helping raise a child–including biological parents, stepparents, foster parents, adoptive parents, grandparents, as well as others household members who take on a significant role in child-rearing.] Yet parents are expected (by themselves and others) to know how to respond appropriately when their child needs attention. What makes this particularly difficult is that not only is there no script for raising children, but that when we're frustrated with our children our reactions often may be driven by experiences with *our* own parents–which may have been less-than-ideal [1]. There is, of course, no shortage of expert advice for how to raise children [2–5]. A central tenet of such guidance is the importance of providing structure and consistency–which helps children better understand their environment and its interconnections, anticipate events and their consequences, and appreciate how these inter-relationships create both expectations and responsibilities [6, 7]. That said, it's also known that both children and parents benefit from developing flexibility and skills for adapting when situations (or our perceptions of them) change [8, 9]. Flexibility of mind (i.e., the ability to switch between different mindsets, tasks, or strategies) [10] promotes resilience by helping us bounce back from a misfortune, recalibrate one's goals and strategies, or carry on in the face of barriers and setbacks [11–13]. For better or worse, raising children offers a great many opportunities to both develop and model such flexibility of mind, to see 'resistance' not as a thing to be beaten down, but as an opportunity for reflection [14]. Toward this end, *humor* could be a valuable parenting tool in part because there are links between humor and cognitive flexibility [15–19]. The study described in this paper is a first step in exploring people's views about humor as a parenting tool–both as children and raising their own children– which could help lay the groundwork for strategies about how to use humor appropriately and productively in times of frustration.

That being said, there is no consensus regarding either what constitutes humor or what grounds it–in part because the term humor is used in various contexts to mean different things [20]. Over the past half-century, a growing body of research has explored the types and structure of humor; [21–24] styles of humor (self-enhancing, affiliative, aggressive, and self-defeating); [25] and correlations between particular types of humor and a variety of outcomes– including subjective wellbeing, emotional regulation, healthy relationships, leadership, teaching, and others [26–33]. Though humor often manifests verbally (jokes, stories, puns, riddles, satire, sarcasm, parody, everyday conversation, etc.) and the written word (e.g., prose, verse, comics), humor can also take a more physical form–paradigmatically, slapstick, but also visual and performing arts [34–37].

Philosophical attempts to characterize humor (sometimes in terms of its progeny, laughter) include works by Plato, Hobbes, Kant, Bergson, and others [19, 38]. At present three of the most commonly referenced theories of humor have emerged from both philosophy and psychology and were intended to encompass all situations of humor and laughter (including inadvertent laughter such as that caused by tickling). Superiority Theory holds that humor and laughter are the result of feelings of superiority over someone else (or the way that we used to be); Incongruity Theory suggests that humor occurs when there is a mismatch between expectations and experience; and Tension and Release Theory, which evolved from Freud's Theory of the Unconscious, suggests that humor results when a buildup of energy created through repression of negative thoughts or discomfort is released through laughter [38]. A more recent theory proposed by behavioral scientist Peter McGraw combines elements of incongruity theory and superiority theory in the Benign Violation Theory, which posits that a joke or a

moment can be perceived as humorous if it is perceived simultaneously as (i) a violation of norms and (ii) benign [39].

On their face, such theories might lead us to view humor as an odd strategy for parenting, and that certain types of humor (especially those involving superiority theory, which includes sarcasm and mockery) are likely to be ineffective if not unhealthy for child-rearing [40]. For example, incongruity-based humor relies on the very opposite of consistency–leveraging surprise, breaking expectations, going the other way [41–43]. So, too, because humor's success is determined in part by how others react, the use of humor by parents could be seen as a 'relinquishing of authority' to their child. Moreover, when done poorly or thoughtlessly, humor can send the wrong message to a child or become a weapon for inflicting harm. That said, humor also has several quite promising features. Notably, humor can induce frameshifts (i.e., changes in perspective) that alter how we interpret an event or response, and thereby open new possibilities for children and parents alike. This includes serving as a distraction that helps children reframe their experience; helping parents create psychological distance (e.g., a moment for perspective-taking) between the immediate experience and their reaction to it; and helping create shared experience that can forge deeper connections amongst family members.

In popular culture, one of the strongest associations between humor and parenting is the "dad joke"–often corny, pun-based, and groan-inducing [44]. But humor's expansive repertoire may have strategic applications for positive parenting techniques akin to how humor has been applied successfully to address various real-world challenges. Tailored improvisation games that use humor and play-based exercises have shown promise for improving the mental health of both individuals with early-stage dementia [45]. and their family caregivers [46]. In the workplace, humor-based play (often cast in terms of *improv*) has been shown to improve attitudes and coping strategies, leading to greater resilience, cohesiveness, and effective leadership [31, 47–50]. So, too, adaptive forms of humor (i.e., affiliative and self-enhancing) are associated with secure attachment [51–56], though with children their developmental stage clearly impacts their ability to appreciate different forms of humor [57–60].

When used appropriately, humor can change the dynamic of situations that are headed for conflict [61–63]. By introducing surprise, incongruity, humor can disrupt established patterns of behavior, which in turn can promote improvisation to yield a different outcome. Imagine, for example, a toddler throwing a full-blown tantrum that continues to escalate despite efforts to calm them. Here, humor as a parenting strategy might have the child's parent declare "OK, it's my turn now," and then dramatically throw their own tantrum. Because this catches the upset toddler by surprise, many children will stop crying and watch the parent's tantrum-performance. As the parent's 'tantrum' winds down, another adult can take their own turn throwing a tantrum. When it becomes the child's turn again, they will typically resume tantruming as if they had never stopped. After a couple of rounds of this, the parent can clap their hands and say "OK, let's play a different game." Playful disruption of this sort not only helps resolve tension, but also models creativity and flexibility of mind, which can serve the interests of both children and parents. Moreover, for parents, the irony of turning a tantrum into a game may provide psychological distance [16] that helps parents relieve their own stress and respond more effectively.

There is also considerable evidence that positive parent-child relationships and greater parental responsiveness to a child's temperament are correlated with greater child well-being [64–66]. There is an expansive literature on the use of play in parenting [67–72].–a related but distinct kind of engagement [73]. So, too, several authors have written about how to promote humor development in children [74–76] and a few small qualitative studies have shown how families of children with disabilities sometimes use humor as an effective coping strategy [77, 78]. But there is little, if any, empirical research on *humor* as a parenting strategy. In fact, in its

more than 30-year existence, the premier journal for humor research (Humor) has not published any articles (conceptual, methods, or outcomes) related to the use of humor in parenting.

There is evidence that people can be taught to use humor more effectively [79, 80]. But it is an open question whether humor (specifically, its uplifting aspects) can be leveraged to help parents respond constructively to challenging situations that might otherwise lead them to respond harshly to their children. As a preliminary step toward answering this question, this pilot study sought to explore whether 18–45-year-olds (putatively, the age range when parents are most likely to be raising young children) consider humor a valuable and viable tool for parenting. To do so, we asked respondents to reflect on their parents' use of humor and how they (i.e., respondents) saw humor as something that could serve a useful role in parenting children.

## Materials and methods

After a review of the literature, we developed a 10-item survey to measure individuals' experiences being raised with humor, as well as their assessment of humor as a potential tool for use in parenting (see Fig 1). The survey was designed using an iterative process that included focus group discussion, construction of question items, field-testing for clarity and face validity, wording revisions, and cognitive interviews to ensure that interpretation of question items corresponded to their intended objectives. Following IRB approval, a REDCap survey was made available from 12/1/21–12/31/21 for up to 400 U.S. participants between the ages of 18–45 using Amazon's Mechanical Turk (MTurk), an online tool that matches eligible participants with public-facing surveys. Consent to participate was implied by individuals proceeding to complete the survey after reading the summary explanation of research. Responses to the 8 quantitative items were measured on a 7-point Likert-type scale and subsequently aggregated into Disagree (*Strongly disagree* and *Disagree*), Indeterminate (*Somewhat disagree*, *Neither agree nor disagree*, and *Somewhat agree*), and Agree (*Agree* and *Strongly agree*). The survey also included 2 open-ended question-items (not included here), and 5 demographic items. All variables were summarized prior to analysis to assess their distributions and determine the best course of analysis. Chi-square tests were used to determine which demographic items were significantly associated with each of the 8 quantitative outcome variables, and to investigate potential associations between 2 of the outcome variables (parent relationship, good job parenting), and the other 6 outcome variables. A significance level of 0.05 was used as the cut-off for statistical significance, and all analyses were performed using SAS version 9.4 (SAS Institute, Cary, NC).

## Results

Of 320 respondents, 312 provided answers to the quantitative items. Most identified as male (63.6%) and White (76.6%), of whom 11.3% were 18–25 years of age, 49.4% were 26–35 years-old, and 39.4% were 36–45 years-old (see Table 1 for full demographics). The majority *Agreed* that they had a positive relationship with the people who raised them (80.1%); that the people who raised them did a good job (64.7%); that they would raise (or already are raising) children in the same way they were raised (54.8%); and that the people who raised them used humor in their parenting (55.2%). So, too, the majority of participants *Agreed* that humor could be an effective parenting tool (71.8%) (see Fig 2); that humor as a parenting tool has more potential benefit than harm (63.3%); that they would value a course on how to utilize humor in parenting (69.7%) (see Fig 3); and that they either plan to (or already do) use humor in parenting their own children (61.8%) (see Fig 4).

| Question | Strongly disagree | Disagree | Somewhat Disagree | Neither Agree nor Disagree | Somewhat Agree | Agree | Strongly Agree |
|---|---|---|---|---|---|---|---|
| I have (or had) a good relationship with the people who raised me. | 1 | 2 | 3 | 4 | 5 | 6 | 7 |
| Overall, I would say that the people who raised me did a good job. | 1 | 2 | 3 | 4 | 5 | 6 | 7 |
| I plan to (or already do) raise my children using the same parenting style that the people who raised me used | 1 | 2 | 3 | 4 | 5 | 6 | 7 |
| I was raised by people who used humor in their approach to parenting. | 1 | 2 | 3 | 4 | 5 | 6 | 7 |
| The next several questions will address the use of "humor" in the parenting process. By humor, we mean things that people do to be comical or amusing | | | | | | | |
| I believe that humor can be an effective parenting tool. | 1 | 2 | 3 | 4 | 5 | 6 | 7 |
| When I think about raising my own children, I plan to (or already do) use humor in my parenting. | 1 | 2 | 3 | 4 | 5 | 6 | 7 |
| Using humor as a tool for parenting has more potential benefit than harm. | 1 | 2 | 3 | 4 | 5 | 6 | 7 |

**Fig 1. Survey questions.**

Interestingly, there were no statistically significant correlations related to participants' age. The same was true for gender, with the exception that more females reported they use (or plan to use) the same parenting style as their own parents (62.3%) compared to males (50.5%, p = 0.036).

By contrast, we found multiple significant differences related to participants' reports regarding the relationships they have with their own parents (see Table 2), and how good a job their own parents had done raising them (see Table 3). Specifically, the ~80% of participants who reported having a good relationship with their parents were significantly more likely to report that their parents had used humor in raising them (63.0%, $p < .001$) compared to those who provided an indeterminate response (23.5%) or denied having a good relationship with their parents (20.0%). So, too, participants who reported a good relationship with their parents were more likely to believe that humor can be an effective parenting tool (77.1%, $p < .001$) compared to those who provided an indeterminate response (49.0%) or denied having a good relationship (54.6%). Those reporting good relationships with their parents were also more likely to say they plan to (or already do) use humor in parenting their own children (65.2%, p

**Table 1. Demographics.**

|  | n (%) |
|---|---|
| Gender |  |
| Female | 108 (35.7%) |
| Male | 192 (63.6%) |
| Other | 2 (0.7%) |
| Missing | 10 |
| Age, years |  |
| 18–25 | 35 (11.3%) |
| 26–35 | 153 (49.4%) |
| 36–45 | 122 (39.4%) |
| Missing | 2 |
| Race/Ethnicity |  |
| Black | 29 (9.6%) |
| Hispanic | 26 (8.6%) |
| White | 231 (76.5%) |
| Other | 16 (5.3%) |
| Missing | 9 |
| Currently raising a child |  |
| Yes | 166 (54.3%) |
| No | 140 (45.8%) |
| Missing | 6 |
| *n* = 312 |  |

< .001) compared to participants who provided an indeterminate response (50.0%) or denied having a good relationship (36.4%).

The same general pattern was seen with the 64.6% of participants who reported that their own parents did a good job raising them–with such individuals reporting higher rates of their parents having used humor (68.4%, p < .001) compared to participants who provided an indeterminate response (33.0%) or denied having a good relationship with their parents (28.6%). Individuals reporting that their parents did a good job raising them were also more likely to

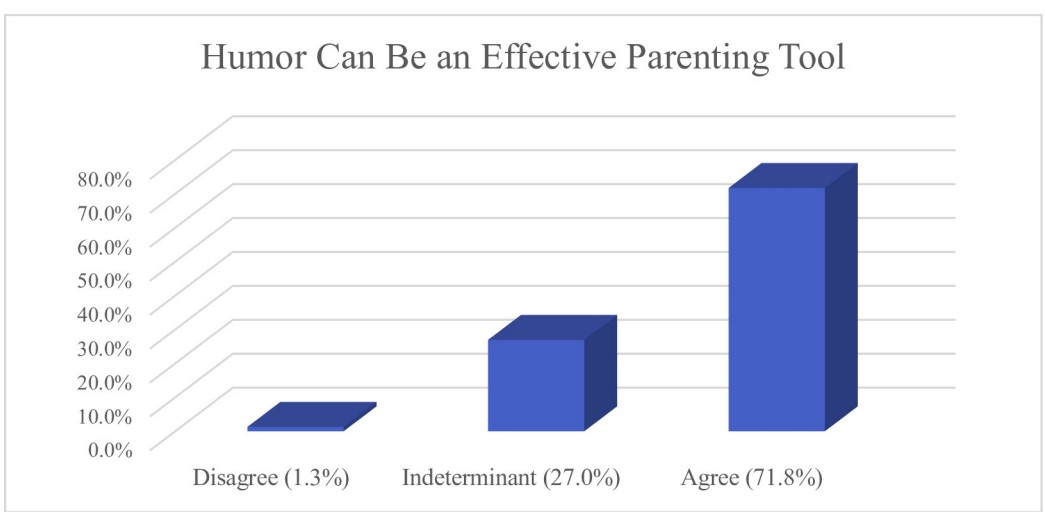

**Fig 2. Views regarding potential efficacy for humor as a parenting tool.**

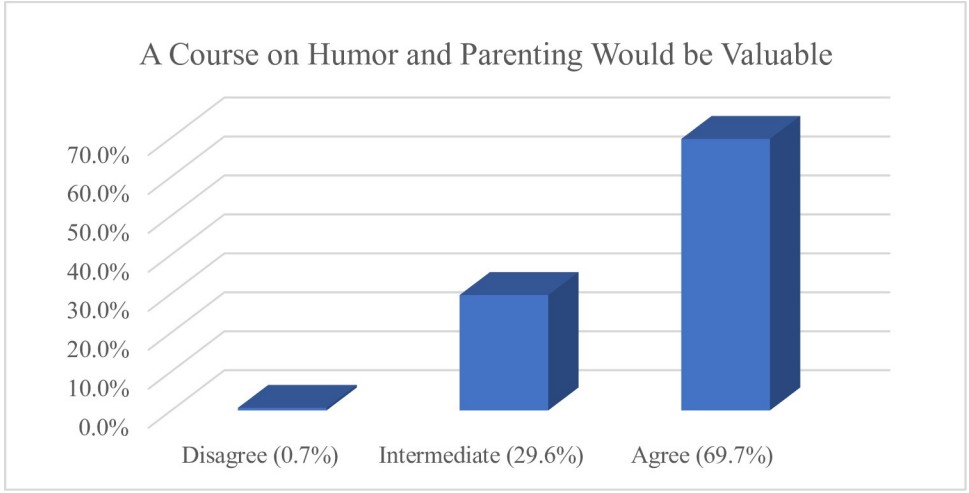

**Fig 3. Views regarding value of a course on humor.**

say that humor can be an effective parenting tool (82.5%, p < .001) compared to those who gave indeterminate responses (52.5%) or denied having a good relationship with their parents (57.1%); were more likely to use (or plan to use) humor in their own parenting (73.9% vs. 39.4% vs. 42.9%, respectively, p < .001); and were more likely to agree that humor as a parenting tool has more potential benefit than harm (72.8% vs. 44.4% vs. 42.9%, respectively, p < .001).

Less surprisingly, but still statistically significant was the finding that participants reporting a good relationship with their own parents were more likely to use (or plan to use) the same

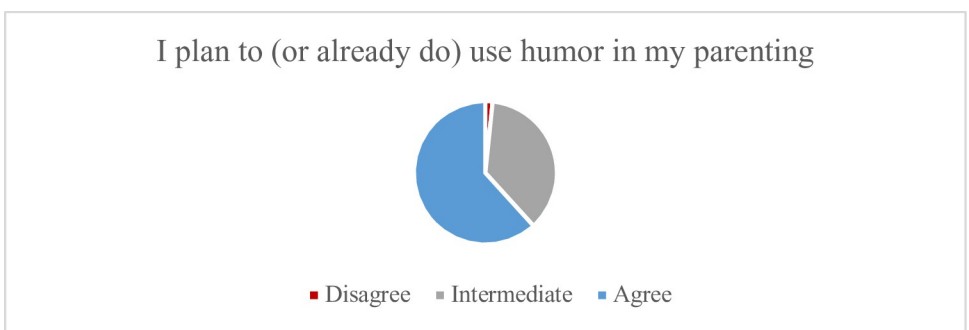

**Fig 4. Views regarding plan to use humor.**

**Table 2. Association between parents' use of humor and relationship with parents.**

|  |  | Parents used humor in their approach to parenting | | |
|---|---|---|---|---|
|  |  | **Disagree** | **Indeterminant** | **Agree** |
| Good relationship with parents | Disagree, N (%) | 5 (50.0%) | 3 (30.0%) | 2 (20.0%) |
|  | Indeterminant, N (%) | 7 (13.7%) | 32 (62.8%) | 12 (23.5%) |
|  | Agree, N (%) | 9 (3.7%) | 82 (33.3%) | 155 (63.0%) |

n = 307[a], *p* < .001 from Chi-square test, Percentages are row percentages

[a] Did not respond = 5

**Table 3. Association between parents' use of humor and parenting.**

| | | Parents used humor in their approach to parenting | | |
|---|---|---|---|---|
| | | **Disagree** | **Indeterminant** | **Agree** |
| Parents did a good job raising me | Disagree, N (%) | 4 (57.1%) | 1 (14.3%) | 2 (28.6%) |
| | Indeterminant, N (%) | 6 (6.0%) | 61 (61.0%) | 33 (33.0%) |
| | Agree, N (%) | 11 (5.6%) | 51 (26.0%) | 134 (68.4%) |

n = 303[a], $p < .001$ from Chi-square test, Percentages are row percentages

[a] Did not respond = 9

parenting style (63.4%, p < .001) compared to those who gave indeterminate responses (22.0%) or denied having a good relationship with their parents (9.1%). A similar association was seen between participants who reported that their parents did a good job raising them, and participants who use (or plan to use) the same parenting style (67.4%, p < .001), compared to participants who gave indeterminate responses (34.3%) or denied having a good relationship with their parents (14.3%).

Worth noting is that fewer than 1% of all respondents reported they did *not* believe that a brief course on how to use humor as a parenting tool would be a valuable resource, whereas 69.7% endorsed such a course, and 29.6% provided an indeterminate response. When comparing respondents currently raising children <18 years old to those who were not, no significant differences (p = .464) were found as to whether they would value such a course on humor. However, individuals currently raising children (vs. those who were not) were more likely to report that humor could be an effective parenting tool (77.0% vs. 65.9%, p = .044), and that humor has more potential benefit than harm (71.1% vs. 53.6%, p = .002). Such differences did not emerge with regard to respondents' race and ethnicity, with the exception of how many people reported that their parents used humor in raising them (61.3% White, 48.3% Black, 30.8% Latino, and 40.0% other, p < .001), and how many planned to use the same parenting style as their own parents (59.7% White, 59.3% Black, 30.8% Latino, and 37.5% other, p = .048).

## Discussion

If the findings of this pilot study are generalizable, they provide preliminary evidence that a significant number of Americans of childbearing/rearing age have had positive experiences being raised with humor and would be interested in learning more about how to use humor as a tool for parenting their own children. Though the use of "play" with young children has an extensive literature, to our knowledge, this is the first study to explore the use of humor in raising children. This is particularly surprising given humor's widespread use and prominence in American culture and well-recognized ability to both bridge differences and help people see things from a new perspective.

Emerging research is demonstrating how humor appreciation is detectable in infants as young as six months of age [81–86]. Over the last 40 years, various authors have discussed the developmental stages of humor as it evolves during childhood [15, 87–89], with some concluding that humor is the inevitable outcome of the progressive evolution of a biological disposition to play [90]. More recently, humor has been recommended as a strategy for early childhood educators [16, 91–93], and there is evidence that humor can be an effective coping strategy for families who have children with disabilities [78, 93–95].

Of course, one must be careful in extrapolating the advantages of humor, as humor is not without its risks–particularly when significant power imbalances exist. So, too, the current

findings do not differentiate the various types of humor or their respective utility, nor do they identify when or how best to leverage humor with children. Nonetheless, the current pilot data suggest that adults of child-raising age are open to the use of humor as one strategy for raising their children and that there is an association between parents' use of humor and subsequent perceptions of one's parents. As such, humor may be an over-looked strategy for helping parents develop, model, and promote cognitive and emotional flexibility. Establishing a foundation that can ground effective strategies for using humor as a parenting tool will require answering basic questions such as:

What characteristics make a situation more or less amenable to using humor as a parenting tool? Which forms of parenting humor are more effective at particular stages of child development and for which goals? What humor-related strategies are better for helping parents and children develop the flexibility of mind for solving problems that might otherwise escalate? How can such skills be taught in a way that minimizes their likelihood of misuse or unintentional harm? Would parents prefer to use humor in place of harsh discipline if it was equally (or more) effective?

## Limitations

The generalizability of these pilot data is limited by several factors, chief among them selection bias. The current findings arise from a relatively small convenience sample that is predominantly white and male. It is also unknown how the views of MTurk respondents compare to the general population. Separately, there are 3 question-items that use parenthetical phrasing to acknowledge that respondents may not yet have children and that respondents' parents may no longer be living. It is possible that respondents interpreted these items as double-barreled questions, though interviews during field-testing did not identify any such confusion. Additionally, it is not clear whether the meaning of 'humor' or 'humor in parenting' was similarly understood by all respondents, nor what factors might affect people's various interpretations of these terms. As noted in the introduction, there is no single agreed-upon definition or interpretation of what constitutes humor. That said, the present study did seek to explore this matter with several qualitative items, but responses to these questions were too meager to offer useful data. Nonetheless, the findings of this pilot study 1) provide a good starting point for conducting a larger, more robust examination of people's experiences and views regarding humor and parenting, and 2) suggest that using humor as a parenting tool may be associated with various beneficial outcomes.

## Conclusion

This pilot study provides preliminary evidence that Americans of childbearing/rearing age have positive views about humor as a parenting tool, and that such use of humor may be associated with various beneficial outcomes. If these findings are generalizable, they potentially open the door to a much deeper and broader exploration of how 'parenting humor' functions and how it can be appropriately leveraged to enhance the experiences of both children and their parents. To that end, future research should examine the ways parents currently use different kinds of humor; children's lived experiences with such humor; and how these uses of humor map onto existing knowledge and theory about how humor functions.

## Author Contributions

**Conceptualization:** Lucy Emery, Benjamin H. Levi.

**Data curation:** Lucy Emery, Erik Lehman, Benjamin H. Levi.

**Formal analysis:** Lucy Emery, Erik Lehman, Benjamin H. Levi.

**Investigation:** Lucy Emery, Benjamin H. Levi.

**Methodology:** Lucy Emery, Benjamin H. Levi.

**Project administration:** Lucy Emery, Benjamin H. Levi.

**Supervision:** Benjamin H. Levi.

**Validation:** Lucy Emery.

**Writing – original draft:** Lucy Emery, Anne Libera, Benjamin H. Levi.

**Writing – review & editing:** Lucy Emery, Anne Libera, Erik Lehman, Benjamin H. Levi.

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
