## [Decision Letter · Decision Letter 0]

16 Apr 2024

PONE-D-23-42264Humor in Parenting: Does It Have a Role?PLOS ONE

Dear Dr. Levi,

Thank you for submitting your manuscript to PLOS ONE. After careful consideration, we feel that it has merit but does not fully meet PLOS ONE’s publication criteria as it currently stands. Therefore, we invite you to submit a revised version of the manuscript that addresses the points raised during the review process.

The article, as acknowledged by the reviewers, explores the role of humour in parenting. However, there are several aspects that could be implemented to improve the quality of the work. These aspects concern firstly, as the reviewer points out (2), the use of concepts and references to theory that should be made more explicit.  I recommend the authors to follow the suggestions of both reviewers in a timely manner.

We look forward to receiving your revised manuscript.

Kind regards,

Ramona Bongelli, Ph.D.

Academic Editor

PLOS ONE

Journal Requirements:

Additional Editor Comments:

The article, as acknowledged by the reviewers, explores the role of humour in parenting. However, there are several aspects that could be implemented to improve the quality of the work. These aspects concern firstly, as the reviewer points out (2), the use of concepts and references to theory that should be made more explicit. I recommend the authors to follow the suggestions of both reviewers in a timely manner.

Reviewers' comments:

Reviewer's Responses to Questions

**Comments to the Author**

1. Is the manuscript technically sound, and do the data support the conclusions?

Reviewer #1: Yes

Reviewer #2: No

2. Has the statistical analysis been performed appropriately and rigorously? 

Reviewer #1: Yes

Reviewer #2: Yes

3. Have the authors made all data underlying the findings in their manuscript fully available?

Reviewer #1: Yes

Reviewer #2: Yes

4. Is the manuscript presented in an intelligible fashion and written in standard English?

Reviewer #1: Yes

Reviewer #2: Yes

5. Review Comments to the Author

Reviewer #1: The manuscript brings original research in the field of parent-child interactions and humor in parenting. The research question is well developed and its presentation is clear enough for the reader. Discussion and conclusion are warranted by the results of the research. The manuscript is well-written and has clear and logical structure.

I have only two minor observations for the authors to consider:

1)On pag. 5, lines 99-100, the authors give a philosophical and psychological overview of the main theories on humour, but only Morreall's philosophical work is mentioned (line 106). I would suggest adding Martin's book 'The psychology of humour. An integrative approach' (2018), for a more complete picture.

2)It is not clear which statistical test was used for the results presented in Tables 2 and 3. I assume they are Pearson correlations, but it would be better to make this explicit. Furthermore, the data in Tables 2 and 3 are not easy to interpret compared to what is clearly stated in the text. I would suggest modifying accordingly.

Reviewer #2: The paper provides an interesting exploration of the potential role of humor in parenting. However, there are several areas that need attention to enhance the clarity, structure, and coherence of the paper. My primary concern with this paper is the oversimplification of the term "humor." There is a substantial body of literature that investigates the differentiation of humor types, such as Humor Styles (Martin et al., 2003) and Comic Styles (Ruch et al., 2018). Therefore, in my opinion, this paper lacks the necessary depth and quality to be considered for publication.

In the introduction, I expected the authors to delve deeper into research exploring the relationship between humor and attachment, particularly addressing the challenges young children face in detecting and understanding irony and sarcasm.

Moreover, the introduction could benefit from a clearer statement of the paper's objectives and research questions to guide the reader.

It would be helpful to provide clear definitions or explanations for terms such as "cognitive flexibility," "frameshifts," and "psychological distance" to ensure that the reader understands these concepts in the context of the paper.

The literature review is not comprehensive and do not cover a range of theories and studies related to humor and parenting, especially related to attachment theory. I therefore recommend the authors to cite more literature in this field.

The paper discusses various theories of humor but could provide a more critical analysis of how these theories apply to parenting. For example, the paper mentions that incongruity-based humor contradicts the need for consistency in parenting, but it could delve deeper into how parents can balance humor and consistency effectively.

The paper identifies a significant gap in empirical research on humor as a parenting strategy. However, the paper could strengthen its argument by discussing why this gap exists and why it is important to address it.

The conclusion summarizes the key points of the paper but could be strengthened by revisiting the research questions and summarizing the potential implications of the study's findings. Additionally, the paper could provide more specific recommendations for future research to address the identified gaps and limitations.

6. PLOS authors have the option to publish the peer review history of their article (what does this mean?). If published, this will include your full peer review and any attached files.

Reviewer #1: No

Reviewer #2: No

---

## [Author Response · Author response to Decision Letter 0]

30 May 2024

We appreciate and agree with the reviewers’ recommendations for improving this manuscript and have made efforts to address each concern.

---

## [Decision Letter · Decision Letter 1]

17 Jun 2024

Humor in Parenting: Does It Have a Role?

PONE-D-23-42264R1

Dear Dr. Levi,

We’re pleased to inform you that your manuscript has been judged scientifically suitable for publication and will be formally accepted for publication once it meets all outstanding technical requirements.

Kind regards,

Ramona Bongelli, Ph.D.

Academic Editor

PLOS ONE

Additional Editor Comments (optional):

Reviewers' comments:

Reviewer's Responses to Questions

**Comments to the Author**

1. If the authors have adequately addressed your comments raised in a previous round of review and you feel that this manuscript is now acceptable for publication, you may indicate that here to bypass the “Comments to the Author” section, enter your conflict of interest statement in the “Confidential to Editor” section, and submit your "Accept" recommendation.

Reviewer #2: All comments have been addressed

2. Is the manuscript technically sound, and do the data support the conclusions?

Reviewer #2: Yes

3. Has the statistical analysis been performed appropriately and rigorously? 

Reviewer #2: Yes

4. Have the authors made all data underlying the findings in their manuscript fully available?

Reviewer #2: Yes

5. Is the manuscript presented in an intelligible fashion and written in standard English?

Reviewer #2: Yes

6. Review Comments to the Author

Reviewer #2: Thank you for addeessing the concerns. In my opinion the paper is now ready for publication as there are no more concerns to be addressed

7. PLOS authors have the option to publish the peer review history of their article (what does this mean?). If published, this will include your full peer review and any attached files.

Reviewer #2: No

---

## [Editor Report · Acceptance letter]

25 Jun 2024

PONE-D-23-42264R1 

PLOS ONE

Dear Dr. Levi, 

I'm pleased to inform you that your manuscript has been deemed suitable for publication in PLOS ONE. Congratulations! Your manuscript is now being handed over to our production team.

Kind regards, 

on behalf of

Professor Ramona Bongelli 

Academic Editor

PLOS ONE